# Comparison between the Friedewald, Martin and Sampson Equations and LDL-C Quantification by Ultracentrifugation in a Mexican Population

**DOI:** 10.3390/diagnostics14121241

**Published:** 2024-06-13

**Authors:** Giovanny Fuentevilla-Álvarez, María Elena Soto, José Antonio García Valdivia, Yazmín Estela Torres-Paz, Reyna Sámano, Israel Perez-Torres, Ricardo Gamboa-Ávila, Claudia Huesca-Gómez

**Affiliations:** 1The Department of Endocrinology, Instituto Nacional de Cardiologia Ignacio Chavez, Juan Badiano Numero 1 Col Seccion XVI, Mexico City 14080, Mexico; fuentevilla_alvarez@hotmail.com; 2Research Direction Instituto Nacional de Cardiologia Ignacio Chavez, Juan Badiano Numero 1 Col Seccion XVI, Mexico City 14080, Mexico; mesoto50@hotmail.com; 3Cardiovascular line in American British Cowdary (ABC) Medical Center Sur Sur 136 No. 116 Col. Las Américas, Mexico City 01120, Mexico; 4Phisiology Department, Instituto Nacional de Cardiología Ignacio Chávez, Juan Badiano No. 1. Col. Sección XVI, Mexico City 14080, Mexico; josantgarval@gmail.com (J.A.G.V.); yazminestela@hotmail.com (Y.E.T.-P.); rgamboaa_2000@yahoo.com (R.G.-Á.); 5Coordinación de Nutrición y Bioprogramación, Instituto Nacional de Perinatología, Ciudad de México 11000, Mexico; ssmr0119@hotmail.com.mx; 6Cardiovascular Biomedicine Department, Instituto Nacional de Cardiología Ignacio Chávez, Juan Badiano No. 1. Col. Sección XVI, Mexico City 14080, Mexico; pertorisr@yahoo.com.mx

**Keywords:** LDL-c quantification, ultracentrifugation, Friedewald, Martin and Sampson equations

## Abstract

Low-density lipoprotein cholesterol (LDL-C), which makes up about 70% of the cholesterol in the blood, is critical in the formation of arteriosclerotic plaques, increasing the risk of heart disease. LDL-C levels are estimated using Friedewald, Martin and Sampson equations, though they have limitations with high triglycerides. Our aim is to compare the effectiveness of these equations versus the ultracentrifugation technique in individuals with and without dyslipidemia and identify precision. There were 113 participants, 59 healthy controls and 54 dyslipidemic patients. Samples were collected after fasting. LDL-C was estimated using the Friedewald, Martin and Sampson equations. The purified LDL-C, ultracentrifugated and dialysized control group without dyslipidemia vs. patients with coronary artery disease (CAD) showed differences in age, HDL-C, triglycerides and glucose non-HDL-C (*p* = 0.001 in all). There were correlations in CGWD between ultracentrifugation and Sampson R-squared (R^2^) = 0.791. In the dyslipidemia control group, ultracentrifugation and Friedewald R^2^ = 0.911. In patients with CAD, correlation between ultracentrifugation and Sampson R^2^ = 0.892; Bland–Altman confirmed agreement in controls without dyslipidemia. The Martin and Sampson equations are interchangeable with ultracentrifugation. Conclusion: The role of LDL analysis using precise techniques is necessary to obtain better control of disease outcomes after the use of precise therapies and suggests verifying its importance through clinical trials.

## 1. Introduction

Low-density lipoprotein cholesterol (LDL-C) consists of molecules that carry most of the cholesterol in blood, nearly 70% [1]. The elevation of LDL-C is harmful because it favors its deposition, oxidation and subsequent inflammatory response that leads to accumulation in the form of arteriosclerotic plaque in the vascular endothelium [2,3]. Keeping triglyceride and cholesterol levels under supervision in patients with dyslipidemia is of vital importance to prevent the development of arteriosclerotic disease, and especially in those who already have the latter because they have an increased risk of cardiac arrest [3]. 

To find out the levels of triglycerides, HDL and LDL cholesterol in the clinic, routine clinical laboratory studies are carried out, the first two being quantified while the last one is estimated through formulas such as Friedewald, Martin and Sampson [4]. Few laboratories carry out the quantification by direct methods of LDL-C, which is replaced with the estimation by equations. In Mexico, the clinical practice guidelines of the main health institutes mark the use of the Friedewald equation for the estimation of LDL-C as ideal [5,6].

The use of the Friedewald equation is recommended at the national level, of obligatory observance, in NOM-037-SSA2-2012 [6]. However, this equation has several limitations: it is not capable of estimating LDL-C levels when triglycerides exceed 4.52 mmol/L [7]. 

Martin et al. developed an equation that modifies the triglyceride-VLDL-C ratio by making the fixed Friedewald ratio adjustable according to triglyceride and non-HDL-C conditions [8]. As the latest advance in equations development to estimate LDL-C, Sampson et al. designed an equation published in 2020. This novel proposal maintains a similar correlation with the estimates of LDL-C obtained with the Martin equation and allows it to be estimated beyond the triglyceride limits of the two previous equations, moving the triglyceride limit up to 9.04 mmol/L [9].

Multiple studies have been performed to validate the equations in various populations [10,11,12]. The rationale for the comparisons can be summarized as comparing the performance of the estimates generated by each equation with a gold standard, either quantification by direct methods using autoanalyzer kits or, to a lesser extent, sequential ultracentrifugation, as it is more expensive in material resources as well as in the performance of samples obtained per unit of time [6].

LDL-C is a key risk factor in the development of atherosclerotic cardiovascular disease and the main therapeutic target in currently available lipid-lowering therapies. The relevance of this risk factor in dyslipidemia is high, and for this reason scientific societies have developed clinical practice guidelines that include as their objective a correct and accurate determination of LDL-C levels in patients with dyslipidemia. In which therapeutic decisions must change according to whether the levels of LDL-C intensify or decrease. LDL-C evaluated using precise technique [13,14]. For all this, the correct determination of LDL-C by different recommended methods is essential.

The objective of our study is to evaluate and determine, in control patients and patients with dyslipidemia, which equation has a better performance when compared with the quantification of isolated LDL-C by sequential ultracentrifugation and by any direct method.

## 2. Materials and Methods

### 2.1. Research Population

A case-control study was carried out with 113 subjects (59 controls and 54 CAD patients) recruited at Instituto Nacional de Cardiología Ignacio Chávez. Inclusion criteria for both groups were as follows: any gender; age > 18 years; Mexican by birth with at least three previous generations of Mexican origin; under 70 years old; and agreeing to participate in the protocol by signing an informed consent. Patients with coronary artery disease (CAD) included in this study had angiographically confirmed obstructive CAD and underwent elective primary coronary artery bypass grafting. This condition is marked by angina symptoms induced by stress or exercise, resulting from significant artery narrowing: ≥50% in the left main trunk or ≥70% in one or more major coronary arteries. The control group was carefully chosen to exclude any comorbidities. These controls were healthy, asymptomatic individuals without a family history of type 2 diabetes, hypertension, or premature cardiovascular disease. They were recruited from blood bank donors and through leaflets posted in social service centers. To confirm the absence of atheroma or subclinical atherosclerosis in the control group, carotid intima-media thickness (cIMT) was measured using ultrasound. Exclusion criteria for both CAD patients and controls included liver disease, kidney disease, cancer, untreated thyroid dysfunction, infectious diseases and corticosteroid treatment. Additionally, the study included only individuals of Mexican descent, and samples were excluded if they were contaminated or insufficient. Study protocol 18-1075 has been approved by the Institute’s Research and Ethics Committee, considering the ethical principles of medical research involving humans as set out in the Declaration of Helsinki and revised by the Tokyo Congress, Japan.

### 2.2. Blood Sampling

Blood samples were obtained by venipuncture after a 12-h fast in ethylenediaminetetraacetate (EDTA-Na) tubes for plasma. Plasma was immediately separated by centrifugation to determine lipid profiles of total cholesterol (TC), triglycerides, high-density lipoprotein cholesterol (HDL-C), low-density lipoprotein cholesterol (LDL-C), and glucose.

### 2.3. Laboratory Analysis

Glucose, TC, and triglycerides were analyzed by enzymatic colorimetry (Roche-Syntex/Boehringer Mannheim, Mannheim, Germany). HDL-C was measured after precipitation of low- and very low-density lipoproteins by phosphotungstate/Mg^2+^ (Roche-Syntex) and LDL-C was estimated by Friedewald, Martin and Sampson equations. For the estimation of LDL-C, the following equations will be used:$$\mathrm{Friedewald}:\;C_{LDL-C}=C_{TC}-C_{HDL-C}-\frac{C_{TG}}{5}$$
$$\mathrm{Martin}:\;C_{LDL-C}=C_{TC}-C_{HDL-C}-\frac{C_{TG}}{Factor\;ajustable}$$
$$\mathrm{Sampson}:\;C_{LDL-C}=\frac{C_{TC}}{0.948}-\frac{C_{HDL-C}}{0.971}-[\frac{C_{TG}}{8.56}+(\frac{C_{TG*}C_{NO\;HDL-C}}{2140})-\frac{C_{TG}^{2}}{16100}]-9.44$$

### 2.4. Dyslipidemia Definition

HDL cholesterol concentration was abnormal if it was ≤0.9 mmol/L. Total cholesterol ≥ 5.17 mmol/L was hypercholesterolemia and triglycerides were considered abnormal at ≥1.69 mmol/L [15]. Dyslipidemia was defined if an individual met at least one of the following criteria: total cholesterol (TC) ≥ 5.18 mmol/L, HDL cholesterol (HDL-C) ≤ 0.91 mmol/L, or triglycerides (TG) ≥ 1.69 mmol/L [6].

### 2.5. Sample Preparation

Initially, plasma samples were thawed at room temperature and centrifuged at 4000 revolutions per minute (rpm) for 5 min to remove precipitates and cells, ensuring clear plasma for subsequent stages of the process.

### 2.6. Ultracentrifugation Phase I: Removal of Chylomicrons, Very Low-Density Lipoproteins (VLDL), and Intermediate-Density Lipoproteins (IDL)

To prepare for ultracentrifugation, the laboratory temperature was adjusted to 20 °C, and the rotor was cooled before use. The process was initiated by placing 1000 µL of plasma into 3.2 mL polycarbonate tubes, increasing its density to 1.019 g per milliliter (g/mL) by adding 80 µL of a 1.21 g/mL potassium bromide (KBr) solution. After adjusting the volume to 3 mL with more KBr solution, the tubes were balanced by weight and centrifuged at 95,000 rpm for 3 h and 15 min at 10 °C, separating the VLDL and IDL lipoproteins. Ultracentrifugation was performed in Optima Max-XP ultracentrifuge, Manufactured by Beckman Coulter, Inc. In Brea, California, United States.

### 2.7. Isolation of LDL

After the first centrifugation, the upper fraction was carefully removed using aspiration techniques to avoid contamination of the LDL. The LDL was collected and resuspended, adjusting the density to 1.063 g/mL with the KBr solution and repeating the centrifugation process under the same conditions to further purify the LDL fraction.

### 2.8. LDL Wash and Dialysis

In the sample preparation phase for LDL dialysis, we commenced by cutting fourteen 7 cm strips from a dialysis membrane, ensuring all handling was done with gloves that had been thoroughly rinsed in 95% alcohol to prevent contamination. Each membrane strip was then soaked in a 1-L beaker filled with 500 mL of distilled water and agitated using a magnetic stirrer for 15 min to equilibrate the membrane. The water was subsequently discarded and replaced with a fresh 500 mL of distilled water, and the process was repeated to ensure thorough rinsing of the membranes.

Once prepared, the membranes were carefully labeled corresponding to the LDL samples they were to carry. For the dialysis process itself, 2 L of phosphate buffer at a neutral pH of 7.4 was prepared in a 3-L beaker. The membranes were then individually removed from the water, gently dried with paper towels, and one end was secured with a specially labeled clamp. The LDL sample was loaded onto the membrane, which was then sealed at the other end with another clamp, ensuring the sample remained contained within the strip during dialysis.

The loaded membranes were placed in the phosphate buffer within the 3-L beaker. This assembly was set to stir gently on a magnetic stirrer inside a refrigeration unit maintained at 4 °C. The first dialysis was conducted over four hours, after which the buffer was refreshed with another 2000 mL and the dialysis continued for an extended period of 12–15 h to achieve optimal exchange conditions. Following this, a final buffer change and a further 4-h dialysis period were completed.

Post-dialysis, each membrane was opened by removing one of the clips, and the dialyzed LDL samples were carefully transferred into labeled 1.5–2 mL Eppendorf tubes. The samples were then immediately stored at 4 °C to maintain their integrity until further analysis. This methodical approach ensures that the LDL samples were adequately prepared, minimizing protein degradation and potential contamination throughout the dialysis process.

### 2.9. LDL Quantitation

Dialyzed LDL cholesterol levels were measured using an enzymatic colorimetric method on a respons 910 autoanalyzer from DiaSys Diagnostic Systems GmbH Manufactured in Holzheim, Germany.

### 2.10. Statistical Analysis

Data were analyzed in the SPSS version 22 program (SPSS Inc., Chicago, IL, USA). In the study, we conducted a comprehensive analysis using anthropometric and biochemical parameters, as well as LDL-C estimations and quantifications. The initial step involved verifying the normality of continuous quantitative variables, which exhibited a non-parametric distribution. Subsequent analyses were performed using the Mann–Whitney U test to compare dyslipidemic groups and those with coronary artery disease (CAD) against control groups. The variables analyzed included age, BMI, and concentrations of total cholesterol, HDL-C, non-HDL-C, triglycerides, and glucose.

Further comparisons were carried out for LDL-C estimations and quantifications among the different groups. Additionally, the Mann–Whitney U test was employed to compare the quantification of LDL-C in the groups against the results obtained from each estimation equation.

To determine the statistical significance of the differences observed between the results from ultracentrifugation and each equation’s estimations across all groups, we conducted one-sample *t*-tests for each comparison, totaling nine tests. This approach helped verify the accuracy and statistical relevance of the differences.

The normality of the results from LDL-C quantifications and estimations was approached to facilitate the creation of Bland–Altman plots, which were used to assess the agreement between the different methods based on the *t*-test differences. Furthermore, scatter plots were generated to visually represent the results from ultracentrifugation and the estimations for all groups. These plots included the computation of correlation coefficients and the root mean square error to evaluate the correlation and precision of the methodologies applied. This comprehensive analysis ensures a robust evaluation of the LDL-C measurement techniques in the context of clinical research. The p values were obtained according to the number of comparisons performed, and it was considered statistically significant if the *p* value was <0.05. 

### 2.11. Ethics Committee

The project obtained approval from the INC’s ethics committee, No. protocol INCAR:18-1075.

## 3. Results

### 3.1. Characteristics of the Study Population

A total of 113 Mexican individuals were studied at the Instituto Nacional de Cardiología Ignacio Chávez, all of whom met the inclusion criteria, of which 59 (52.2%) were considered the control group and 54 (47.8%) were considered the coronary artery disease (CAD) group.

Table 1 shows the main anthropometric and biochemical characteristics of the study population stratified into controls and patients. The group of patients presented significant differences with respect to the control in age (*p* < 0.001), BMI (*p* = 0.015), TC (*p* < 0.001), HDL-C (*p* < 0.001), non-HDL-C (*p* < 0.001), LDL-C (*p* < 0.001) and glucose (*p* < 0.001).

It was necessary to stratify the two study groups considering dyslipidemia. The control group was divided into dyslipidemic 27 (45.8%) and non-dyslipidemic 32 (54.2%); For the CAD group, 100% of the individuals presented dyslipidemia.

Table 2 shows the main anthropometric and biochemical characteristics of the study population stratified into controls (dyslipidemic and non-dyslipidemic) and the CAD group. The control group without dyslipidemia showed significant differences compared to the dyslipidemic control group: HDL-C (*p* < 0.001), non-HDL-C (*p* = 0.002), and TG (<0.001). When comparing the control group without dyslipidemia against patients with CAD, significant differences were found in age (<0.001), TC (0.001), HDL-C (0.001), non-HDL-C (0.033), TG (< 0.001) and glucose (<0.001).

### 3.2. LDL-C Levels Estimated with the 3 Equations and Quantified after Ultracentrifugation Isolation

Table 3 shows the levels of LDL-C in mmol/L calculated by the three equations and by ultracentrifugation in the three study groups. When comparing the control group without dyslipidemia versus the control group with dyslipidemia, no significant differences were found in the LDL-C levels obtained with each method. However, when comparing data from the control group without dyslipidemia against the dyslipidemia CAD, significant differences were found in plasma LDL-C levels for all methods used: ultracentrifugation (*p* = 0.009), Friedewald (*p* < 0.001), Martin (*p* = 0.001) and Sampson (*p* = 0.001).

### 3.3. Correlation Test

To corroborate if there is a correlation between the data obtained by ultracentrifugation and the equations evaluated in the three study groups (dyslipidemic controls, non-dyslipidemic controls, and dyslipidemic CAD), a Spearman correlation analysis was performed.

Figure 1 shows the correlations of the three study groups. The non-dyslipidemic control group showed that the best linear model was ultracentrifugation vs. Sampson with a root mean square error (RMSE) of 10.262 and an adjusted R-squared (R^2^) of 0.791. In the case of the dyslipidemic control group, it was found that the best model was presented when correlating ultracentrifugation vs. Friedewald, obtaining an RMSE of 8.999 and an adjusted R^2^ of 0.911. Finally, in the group of patients with CAD undergoing revascularization, it was observed that the best correlation was presented by ultracentrifugation vs. Sampson, with an RMSE of 15.858 and an adjusted R^2^ of 0.892.

### 3.4. Concordance Test

It was necessary to perform a Bland–Altman analysis to verify if the data obtained by the different methods are concordant and interchangeable with sufficient precision. In principle, due to the non-parametric distribution of the data, an approximation to the normality of the recorded LDL-C data was made, transforming them to logarithms with natural logarithm (ln). With the data in “ln” it was verified how well the methods agree on average: the differences were made between the ultracentrifugation data and the three equations, in the control groups (dyslipidemic and non-dyslipidemic) and EAC; subsequently, a student’s *t*-test was performed with the differences obtained by ultracentrifugation with each equation, in the three study groups.

Table 4 displays the average agreement results using *p*-values calculated from the student’s *t*-test conducted in each study group. Only in the case of controls without dyslipidemia, when comparing ultracentrifugation vs. Martin and Sampson, did we not obtain significant differences, which means that the methods are concordant. Only Bland–Altman diagrams were performed, which did not show significant differences (Figure 2). The Bland–Altman diagrams show that the comparison of ultracentrifugation with the Martin equation showed a Bland–Altman coefficient B = 0.043 and systematic error d = 0.0229 with agreement limits between −0.1651 and 0.2109. The comparison of ultracentrifugation with the Sampson equation is as follows:

## 4. Discussion

This study aimed to evaluate the performance of the Friedewald equation (widely used in hospital and conventional clinical laboratories) and the experimental Martin and Sampson equations, compared to the quantification of LDL isolated by ultracentrifugation, in Mexican patients.

In recent years, many clinical guidelines have highlighted LDL cholesterol (LDL-C) as the focus for treatment in cardiovascular diseases and for preventing the development of atherosclerosis [16,17]. The significance of LDL-C as a biomarker means that knowing its plasma levels is essential, as they mirror the biological state of the individual. While various equations are available to estimate LDL-C levels, it is crucial to assess their accuracy against a reference method to determine which provides the most reliable estimation [18,19,20].

Results show that individuals with coronary artery disease (CAD) have lower plasma concentrations of LDL cholesterol (LDL-C) compared to the control group without dyslipidemia. This can be attributed to the fact that the patients are undergoing statin treatment [21,22,23]. As previously described, statin use promotes the endocytosis of cholesterol-loaded LDL due to increased expression of LDL receptors (r-LDL), which results from the inhibition of HMG CoA reductase, the rate-limiting enzyme in intracellular cholesterol synthesis. All of this leads to a decrease in LDL-C concentration [24,25].

According to Table 2, the patient group with coronary artery disease (CAD) shows decreased levels of total cholesterol (CT) and HDL cholesterol (C-HDL), and triglycerides (TG) nearing the hypertriglyceridemia threshold of 1.695 mmol/L. Estimating LDL cholesterol (C-LDL) under these conditions tends to result in an underestimation, which is reflected in the results of Table 3, CAD group. Here, the difference in medians between the Friedewald formula and ultracentrifugation is the largest compared to the other two equations, 13 mmol/L lower than what is quantified by ultracentrifugation. A possible explanation for this is that the underestimation of LDL cholesterol (LDL-C) levels using the Friedewald formula is due to its overestimation of VLDL cholesterol (VLDL-C) levels [26].

Statin treatments, as previously described, reduce the synthesis of VLDL-C, which in turn lowers VLDL cholesterol (VLDL-C) levels [27,28]. Therefore, VLDL-C is lower than the value set by the Friedewald equation since it no longer represents 20% of the plasma triglyceride concentration. Even though this occurs, the fixed factor of the Friedewald formula causes LDL cholesterol (LDL-C) estimates in the group treated with statins to be lower than those obtained by ultracentrifugation [29].

The findings from our linear regression analyses reveal differing efficiencies in the applicability of lipid estimation equations across various groups, highlighting the nuanced nature of lipid profiling in clinical settings.

Non-Dyslipidemic Controls:

In the group of non-dyslipidemic controls, the Martin equation demonstrated the best correlation (R^2^ = 0.791 and RMSE = 10.262). This superior performance of the Martin equation over the Sampson and Friedewald equations aligns with previous studies that have shown its accuracy in individuals with triglyceride levels below 4.52 mmol/L. This precision is crucial for effective clinical assessments in non-dyslipidemic populations where accurate lipid profiling can guide preventive strategies against cardiovascular diseases [30].

Dyslipidemic Controls:

Interestingly, in the dyslipidemic control group, the Friedewald equation emerged as the most accurate (R^2^ = 0.911 and RMSE = 8.999), which is contrary to initial expectations based on its historical performance. This anomaly can be explained by the triglyceride levels of the patients, which ranged from 1.6611–2.4747 mmol/L, well below the equation’s upper functional threshold of 4.51 mmol/L [31]. Moreover, the non-HDL cholesterol levels in these patients were above 2.59 mmol/L, confirming the reliability of the Friedewald equation under these specific conditions [12,32]. This demonstrates that despite its general limitations, the Friedewald equation can still provide reliable estimates in certain dyslipidemic populations.

Coronary Artery Disease (CAD) Group:

For the CAD group, the Sampson equation proved most effective (R^2^ = 0.892 and RMSE = 10.858). The Martin equation, while generally robust, tends to lose precision with LDL-C values deviating from 1.813 mmol/L. Conversely, the Sampson equation maintains greater accuracy for LDL-C levels above this threshold, making it particularly suitable for the CAD patient cohort. This group typically presents with higher LDL-C levels due to their disease state, necessitating an equation that can accurately reflect these elevated levels [33]. These findings underscore the importance of selecting appropriate lipid estimation equations based on the specific lipid profile and health condition of the patient group. The superior correlation of the Sampson equation in the CAD group can be attributed to its enhanced precision at higher LDL-C levels, which are more prevalent among individuals with coronary artery disease [12]. This accuracy is critical for managing and monitoring CAD, as lipid-lowering therapy is a cornerstone of treatment in these patients.

Thus, the study supports the use of the Sampson equation in clinical settings involving CAD patients due to its demonstrated reliability in accurately estimating LDL-C levels in contexts where precision is paramount for therapeutic decision-making and risk assessment.

In the Bland–Altman analysis, although initially the Martin and Sampson equations appeared to have similar accuracy to ultracentrifugation, a deeper analysis reveals significant differences. According to Figure 2, while the systematic error of the Martin equation is slightly less than that of Sampson (0.0229 vs. 0.0279), the Bland–Altman coefficient reveals that the slope of the relationship with ultracentrifugation in the Martin equation is positive. This indicates that at higher LDL levels, the difference between the values measured by ultracentrifugation and those estimated by the Martin equation increases, suggesting less concordance compared to the Sampson equation, which has an almost constant slope, indicating a consistent difference regardless of the LDL level [34]. Therefore, the Sampson equation demonstrates better concordance for the non-dyslipidemic control group.

The analysis of LDL-C quantification methods using the Martin, Sampson and Friedewald equations compared to ultracentrifugation reveals significant differences in precision and correlation depending on the lipid conditions of the individuals.

The results indicate that the Martin equation works better in the non-dyslipidemic control group with low triglycerides [35,36], while the Friedewald equation shows a better correlation in dyslipidemics with moderate triglyceride levels, suggesting its validity under certain triglyceride thresholds [36,37].

On the other hand, the Sampson equation proves to be more consistent in the group with coronary artery disease (CAD), adapting better to variations in LDL-C greater than 1.813 mmol/L [38]. However, neither method achieves accuracy in reporting LDL-C in patients with hypertriglyceridemia.

Although extensive efforts have been made in our country to compare these equations in our population [38,39,40], this study is among one of the first in Mexico to compare the direct measurement of LDL-C using the Friedewald, Martin and Sampson equations against the gold standard of ultracentrifugation, specifically in patients with at least three generations of Mexican ancestry. While previous studies in Mexico have compared these equations, our work provides a clearer insight into the lipid profiles typical of the Mexican population and enhances the precision and relevance of our findings.

This study provides a detailed comparative evaluation of the Friedewald, Martin and Sampson equations for estimating LDL-C against quantification by ultracentrifugation, within a cohort of Mexican patients, including both apparently healthy individuals and those with developing coronary artery disease (CAD). Unlike previous studies, our research stands out by conducting this comparison across these two specific groups, thereby offering a more comprehensive insight into the effectiveness of these equations under various clinical conditions. 

Our findings reveal that while the Sampson equation proved most suitable for patients with CAD, the Martin equation was most effective in the non-dyslipidemic control group, and the Friedewald equation was the most accurate in the dyslipidemic control group, provided triglyceride levels remained below 4.52 mmol/L. This study underscores the necessity of carefully selecting the estimation equation based on the patient’s lipid profile and health status.

Ultracentrifugation, although still the gold standard for quantifying LDL-C [41,42], has limitations in terms of feasibility due to its high complexity and cost, which restricts its widespread use in all hospitals. This highlights the importance of having accurate and reliable estimation equations that can be effectively used in routine clinical practice [43].

This study highlights that a single equation is insufficient to accurately quantify LDL-C across the diverse Mexican population, which exhibits significant variability in lipid profiles due to the high rate of cardiovascular diseases such as coronary artery disease (CAD). Our results demonstrate that the appropriateness of each estimation equation varies depending on the specific clinical context of each patient [44,45].

This nuanced understanding underscores the complexity of lipid management in a population with substantial clinical diversity and emphasizes the necessity for personalized approaches in the estimation of LDL-C. Tailoring the choice of equation to individual patient profiles is crucial for achieving accurate and clinically relevant lipid measurements, essential for effective disease management and prevention strategies in Mexico.

This is of particular importance for the physician treating their patients, since a considerable underestimation of the Friedewald and Sampson equation could lead to insufficient treatment in hypertriglyceridemia. It is important to inform physicians of these results, since the doctor who treats these patients must know the existence of all the methods used in the laboratory to determine the levels. Likewise, doctors must have knowledge of the advantages and disadvantages of each method when analyzing a patient in the phase of lipid imbalance. Thus, caution must be exercised in clinical judgement when estimating a patient’s LDL-C in relation to the range of triglycerides and type of method.

## 5. Conclusions 

Ultracentrifugation is a powerful technique for the separation and analysis of particles in biological samples, but its high costs and technical complexities hinder its accessibility and usefulness in most hospitals. Being able to explore the agreement with other alternative techniques allows us to evaluate the power that each one achieves for specific determinations in LDL. A serious limitation in this study is not having included a group with hypertriglyceridemia of genetic origin, which in the clinic is of serious importance, and therefore it is required to evaluate the accuracy of each of these tests in order to indicate the usefulness that the user can achieve with each of them.

## Figures and Tables

**Figure 1 diagnostics-14-01241-f001:**
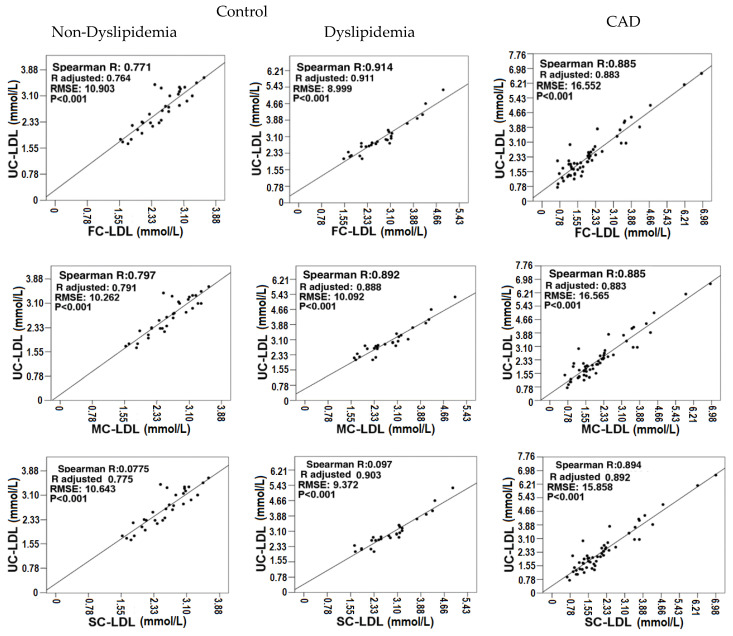
Correlation graphs of ultracentrifugation with the three equations, in both controls and EAC. Spearman R correlation coefficient. Adjusted R: adjusted Spearman correlation coefficient. RMSE: root mean square error. Statistical significance with *p* < 0.05. FC (Friedewald quantification), MC (Martin quantification), SC (Sampson quantification), UC (ultracentrifugation quantification).

**Figure 2 diagnostics-14-01241-f002:**
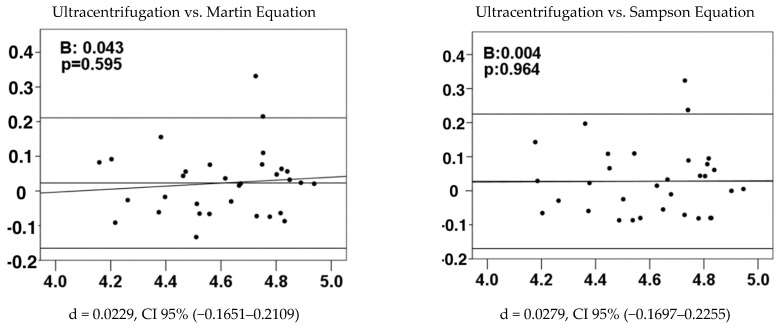
Bland–Altman graphs of non-dyslipidemic patients. d = systematic error between both methods. 95%CI: limits of agreement with 95% confidence. B: Bland–Altman coefficient. *p*: statistical significance with *p* < 0.05.

**Table 1 diagnostics-14-01241-t001:** Biochemical and anthropometric parameters.

Variable	Controls n = 59	CAD n = 54	*p*
Sex (%M/%F)	55.9%/44.1%	88.9%/11.1%	-
Age (years)	47 (44–51)	62 (57–69)	<0.001
BMI (kg/m^2^)	27.7 (25.4–29.8)	26.2 (24.4–27.8)	0.015
TC (mmol/L)	4.50 (3.98–5.02)	3.37 (2.80–4.22)	<0.001
HDL-C (mmol/L)	1.09 (0.98–1.29)	0.85 (0.70–0.93)	<0.001
Non-HDL-C (mmol/L)	3.29 (2.90–3.91)	2.69 (1.99–3.31)	<0.001
LDL-C (mmol/L)	2.82 (1.68–5.31)	2.07 (1.53–2.90)	<0.001
Triglycerides (mmol/L)	1.33 (1.05–1.95)	1.47 (1.17–2.13)	0.300
Glucose (mmol/L)	5.27 (4.8–5.49)	5.661 (5.21–6.77)	<0.001
Statins (%)	0%	100%	-
Smoking (%)	10.5%	17.1%	-
Alcoholism (%)	0%	2.4%	-

%M/%F: percentage of male individuals/percentage of female individuals. BMI: body mass index, TC: total cholesterol, HDL-C: high-density lipoprotein cholesterol, non-HDL-C: non-high-density lipoprotein cholesterol, LDL-C: low-density lipoprotein cholesterol, TG: triglycerides. Values expressed as median (1st quartile–3rd quartile). Statistical significance at *p* < 0.05. Statistical test: Mann–Whitney U.

**Table 2 diagnostics-14-01241-t002:** Anthropometric and biochemical parameters of controls and CAD.

Variable	Controls (n = 59)		CAD Dyslipidemia (n = 54)	p1	p2
	Non-Dyslipidemia (n = 32)	Dyslipidemia (n = 27)			
Sex (%M/%F)	53.3%/43.7%	60.7%/39.3%	88.9%/11.1%	-	-
Age (years)	48 (44–53)	47 (44–50)	62 (57–69)	0.393	<0.001 *
BMI (kg/m^2^)	26.8 (25.4–28.7)	28.9 (26.1–31.3)	26.0 (24.1–27.7)	0.055	0.138
TC (mmol/L)	4.40 (3.96–4.71)	4.76 (4.04–5.46)	3.37 (2.67–4.27)	0.059	<0.001 *
HDL-C (mmol/L)	1.22 (1.04–1.40)	0.96 (0.85–1.16)	0.78 (0.67–0.90)	<0.001 *	<0.001 *
Non-HDL-C (mmol/L)	3.16 (2.56–3.57)	3.73 (3.21–4.24)	2.68 (1.94–3.39)	0.002 *	0.033 *
Triglycerides (mmol/L)	1.12 (0.92–1.33)	2.15 (1.66–2.48)	1.67 (1.29–2.31)	<0.001 *	<0.001 *
Glucose (mmol/L)	5.27 (4.88–5.49)	5.66 (5.21–6.77)	5.85 (5.28–7.89)	0.831	<0.001 *
Statins (%)	-	-	100%	-	-

%M/%F: percentage of male individuals/percentage of female individuals. BMI: body mass index, TC: total cholesterol, HDL-C: high-density lipoprotein cholesterol, non-HDL-C: non-high-density lipoprotein cholesterol, TG: triglycerides. Values expressed as median (1st quartile–3rd quartile). Statistical significance at *p* < 0.05. Statistical test: Mann–Whitney U. *: *p*-value less than 0.05, indicating statistically significant difference. p1: controls non-dyslipidemia vs. controls with dyslipidemia. p2: controls non-dyslipidemia vs. CAD dyslipidemia.

**Table 3 diagnostics-14-01241-t003:** LDL-C levels by method, in both controls and CAD.

Method (mmol/L)	Controls (n = 59)		CAD Dyslipidemia (n = 54)	p1	p2
	Non-Dyslipidemia (n = 32)	Dyslipidemia (n = 27)			
Ultracentrifugation	2.72 (2.25–3.21)	2.82 (2.64–3.35)	2.07 (1.53–2.90)	0.206	0.009 *
Friedewald	2.61 (2.09–3.01)	2.67 (2.21–3.17)	1.74 (1.24–2.35)	0.438	<0.001 *
Martin	2.61 (2.21–3.13)	2.68 (2.35–3.46)	1.85 (1.40–2.51)	0.632	0.001 *
Sampson	2.61 (2.11–3.09)	2.86 (2.36–3.50)	1.82 (1.33–2.45)	0.193	0.001 *

CAD: Coronary Arterial Disease. Values expressed as median (1st quartile–3rd quartile). Statistical significance at *p* < 0.05. Statistical test: Mann–Whitney U. *: *p*-value less than 0.05, indicating statistically significant difference. p1: controls non-dyslipidemia vs. controls with dyslipidemia. p2: controls non-dyslipidemia vs. CAD dyslipidemia.

**Table 4 diagnostics-14-01241-t004:** Concordance test between ultracentrifugation and the Friedewald, Martin and Sampson equations in the three groups.

Equation	Ultracentrifugation		
	Control (n = 59)		CAD Dyslipidemia (n = 54)
	Dyslipidemia (n = 32)	Non-Dyslipidemia (n = 27)	
Friedewald	0.023	<0.001 *	<0.001 *
Martin	0.187	0.002 *	<0.001 *
Sampson	0.128	0.032 *	<0.001 *

CAD: coronary artery disease. Statistical test: one-variable student *t*. *: *p* with a value less than 0.05, there is a statistically significant difference.

## Data Availability

Due to confidentiality agreements, the data underlying this study are not publicly available. Access to the data can be requested through c_huesca@yahoo.com following their confidentiality protocols.

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
