# Peer review of "Comparison between the Friedewald, Martin and Sampson Equations and LDL-C Quantification by Ultracentrifugation in a Mexican Population"

_diagnostics, 2024, doi:10.3390/diagnostics14121241_

Round 1

Reviewer 1 Report

Comments and Suggestions for Authors

It is a well-designed study. Just a few comments:

- Make a better conclusion for the ultracentrifugation method.

- Highlight the novelty of this work. I have some concerns about it.

- Add this reference: doi.org/10.1016/j.heliyon.2023.e21875 as a study that indicates higher levels of LDL should be calculated with Friedewald and lower than 70 levels by Martin.

Author Response

24 may 2024

Reviewer 1

We appreciate your comments regarding our work. All authors appreciate the suggestions and changes you make, as they occasionally improve the writing.

It is a well-designed study. Just a few comments:

1.- Make a better conclusion for the ultracentrifugation method.

1.- Answer  Thanks for the comment, we have added the following to the text in the following lines

This study provides a detailed comparative evaluation of the Friedewald, Martin, and Sampson equations for estimating LDL-C against the quantification by ultracentrifugation, within a cohort of Mexican patients, including both apparently healthy individuals and those with developing coronary artery disease (CAD). Unlike previous studies, our research stands out by conducting this comparison across these two specific groups, thereby offering a more comprehensive insight into the effectiveness of these equations under various clinical conditions.

Our findings reveal that while the Sampson equation proved most suitable for patients with CAD, the Martin equation was most effective in the non-dyslipidemic control group, and the Friedewald equation was the most accurate in the dyslipidemic control group, provided triglyceride levels remained below 400 mg/dL. This study underscores the necessity of carefully selecting the estimation equation based on the patient's lipid profile and health status.

Ultracentrifugation, although still the gold standard for quantifying LDL-C, has limitations in terms of feasibility due to its high complexity and cost, which restricts its widespread use in all hospitals. This highlights the importance of having accurate and reliable estimation equations that can be effectively used in routine clinical practice.

2.- Highlight the novelty of this work. I have some concerns about it.

2.- Answer We have added to the text and briefly highlighted the importance of the work, we appreciate your suggestion. Lines

This study highlights that a single equation is insufficient to accurately quantify LDL-C across the diverse Mexican population, which exhibits significant variability in lipid profiles due to the high rate of cardiovascular diseases such as coronary artery disease (CAD). Our results demonstrate that the appropriateness of each estimation equation varies depending on the specific clinical context of each patient.

This nuanced understanding underscores the complexity of lipid management in a population with substantial clinical diversity and emphasizes the necessity for personalized approaches in the estimation of LDL-C. Tailoring the choice of equation to individual patient profiles is crucial for achieving accurate and clinically relevant lipid measurements, essential for effective disease management and prevention strategies in Mexico.

3.- Add this reference: doi.org/10.1016/j.heliyon.2023.e21875 as a study that indicates higher levels of LDL should be calculated with Friedewald and lower than 70 levels by Martin.

3.- Answer  Thanks for the suggestion we have added the reference in the discussion

Reviewer 2 Report

Comments and Suggestions for Authors

The authors examined the validity of LDL-c estimation equations for Mexican populations. The results seem helpful. However, similar studies were made by several researchers. The authors should cite these studies and compare them with their results.

Furthermore, the number of samples seems to be too small.

The authors cited several non-English language references. I do not see the necessity. English language references are more convenient for ordinary readers.

More careful bibliographic studies are necessary. 

Author Response

24- may-2024

Reviewer 2

We thank the reviewer for the comments and suggestions made towards this work, we have taken all his suggestions into account and added them to the text.

1.-The authors examined the validity of LDL-c estimation equations for Mexican populations. The results seem helpful. However, similar studies were made by several researchers. The authors should cite these studies and compare them with their results.

1.- Answer  We have added bibliography concerning these topics and we have modified the discussion. We appreciate your comments.

2.-Furthermore, the number of samples seems to be too small.

We appreciate the reviewer's observation.

  1. Answer The sample is small because although we calculated the sample size based on the article published in “Tsai MY, Georgopoulos A, Otvos JD, Ordovas JM, Hanson NQ, Peacock JM, Arnett DK. Comparison of ultracentrifugation and nuclear magnetic resonance spectroscopy in quantifying triglyceride-rich lipoproteins after an oral fat load. Clin Chem. 2004 Jul;50(7):1201-4. doi: 10.1373/clinchem.2004.032938. Epub 2004 May 13. PMID: 15142979.”

      In which the correlation achieved by the ultracentrifugation test is 0.90 with RME. This test is the gold standard, and we compare it with the other tests, considering that they reach a lower power than the gold standard test. We calculate that it is 0.85 in them. 

Therefore, when we calculated the required sample using Fisher's z test to compare two correlation coefficients, we found that we needed 120 subjects in total; however, we only reached 113.

Our power analysis was conducted with meticulous care, utilizing the formula Power Estimation and Sample Size for Correlation Coefficients as recommended by Bernard Rosner (Fundamentals of Biostatistics Fifth Edition)ø(√(n-3 Z0-Z1-α)) , and specifying the alternative hypothesis test. This comprehensive approach should instill confidence in the validity of our findings.

Which gave us a power in our sample of this study of 0.78.

We only have a minimum deficit left than necessary.

3.-The authors cited several non-English language references. I do not see the necessity. English language references are more convenient for ordinary readers.

3.- Answer  We have corrected this in the text, thanks for the suggestion

4.- More careful bibliographic studies are necessary.

4.-R Answer We have corrected this in the text, thanks for the suggestion
